# Concentration of Nutrients in Leaves, Flowers, and Fruits of Genotypes of *Coffea canephora*

**DOI:** 10.3390/plants10122661

**Published:** 2021-12-03

**Authors:** Millena Monteiro dos Santos, Cleidson Alves da Silva, Eduardo France Oza, Ivoney Gontijo, José Francisco Teixeira do Amaral, Fábio Luiz Partelli

**Affiliations:** 1Programa de Pós-Graduação em Agricultura Tropical, Universidade Federal do Espírito Santo, São Mateus 29932-540, Espírito Santo, Brazil; millena_monteiro@hotmail.com; 2Programa de Pós-Graduação em Fitotecnia, Universidade Federal de Lavras, Lavras 37200-000, Minas Gerais, Brazil; cleidsonsilva.cas@gmail.com; 3Programa de Pós-Graduação em Ecologia e Recursos Naturais, Universidade Estadual do Norte Fluminense Darcy Ribeiro, Campos dos Goytacazes 28013-602, Rio de Janeiro, Brazil; eduardo.franceoza@hotmail.com; 4Departamento de Ciências Agrárias e Biológicas, Universidade Federal do Espírito Santo, São Mateus 29932-540, Espírito Santo, Brazil; ivoneygontijo@yahoo.com.br; 5Departamento de Engenharia Rural, Centro de Ciências Agrárias e Engenharias, Universidade Federal do Espírito Santo, Alegre 29500-000, Espírito Santo, Brazil; jftamaral@yahoo.com.br

**Keywords:** conilon coffee, genetic improvement, nutrient absorption curve, mineral nutrition

## Abstract

It is important to consider the nutritional demand among genotypes of the same species to achieve high yields. Thus, the objective of this study was to determine the concentration of nutrients in leaves, flowers, grains, and fruit straw in conilon coffee genotypes. The experiment was carried out under a randomized block design, with three replications and the evaluation of nine genotypes. Leaf collections were carried out every three months, from August 2019 to May 2020. Flowers were collected at flowering in July 2019 and fruits in June 2020, upon complete maturation of the genotypes. The materials were dried in an oven and sent for laboratory analysis to determine the nutritional content. Data were subjected to analysis of variance followed by a comparison of means and estimates of genetic parameters and clustering using the hierarchical method (UPGMA). The nutrients found in the highest concentrations in the evaluated plant organs were N and K for macronutrients and Fe for micronutrients. For the leaves, the concentrations of the main nutrients were high in the first and reduced in the last evaluated periods, possibly due to mobilization to the fruits. Considering all the plant tissues evaluated, the order of concentration of macronutrients and micronutrients was N > K > Ca > P = Mg = S and Fe > B > Mn > Cu > Zn, respectively. For a nutritional diagnosis, it is important to take comparisons of the genetic diversity and evaluation periods into consideration.

## 1. Introduction

High yields in coffee crops have been achieved using superior genotypes developed over decades of genetic improvement [1]. *Coffea canephora* Pierre ex A. Froehner is a highly heterozygous species, and distinct characteristics are expressed in the different genotypes [2]. Among them, we can mention drought tolerance [2], accumulation of biomass and nutrients [3,4], vegetative growth [5], and absorption and efficiency of nutrient use [6,7]. This broad diversity enables the selection and introduction of superior genotypes into the production chain.

Highly productive genotypes have high nutritional requirements [8], and they may or may not vary between strains. The length of the reproductive cycle [3,4], anatomy of the plant organ [6], and phenological phases [9] are some examples of mechanisms responsible for the nutritional variation in coffee genotypes [10]. They can vary in concentration and have a greater requirement of one or more elements [11,12,13].

In general, in *C. canephora* plants, nitrogen (N), calcium (Ca), and potassium (K) are absorbed in greater amounts, while phosphorus (P), magnesium (Mg), and sulfur (S) are required less [4,14,15]. In leaves, there is a decreased concentration of macronutrients considered mobile in the plant during the reproductive phase. This decrease results from the migration of photoassimilates and mineral elements to drains, such as flowers and fruits, after the degradation of cellular compounds [16]. In several varieties of conilon coffee, Oliosi et al. [10] and Covre et al. [15] observed greater demand in the following order: N, P, K, Ca, S, Mg, B, Fe, Mn, Zn, Cu in the reproductive stages, in anthesis, pre-flowering, grain filling, and fruiting.

The plant’s nutritional imbalance can promote the excessive accumulation of elements [10], triggering a series of undesirable metabolic processes, which can vary according to the type of organ in the plant and may have an influence on the quality of the beverage [17]. The plant’s nutritional status is an essential tool in managing coffee production, determined by the nutritional content in different plant tissues. Although the evaluations of organs from coffee trees at different phenological stages are essential tools to determine the proper fertilization management, the existing reference values of the critical level and sufficiency range for nutritional diagnosis do not consider the genetic diversity among genotypes [8].

Several plant organs can provide subsidies for adequate nutritional management. Flower analysis enables us to advance in nutritional corrections during vegetative growth, avoiding further losses [18,19]. The grains and straw analysis provides essential answers about calculating the nutrients responsible for crop productivity [15]. The most used organ of the coffee plant in these analyses is the leaves. They are physiologically active organs, present in all phenological cycles of the crop, favoring the monitoring of the plant’s nutritional status [18,19].

Therefore, the objective of this study was to determine the genetic variability for the concentration of nutrients in different plant organs throughout the reproductive cycle of nine genotypes of conilon coffee (*Coffea canephora*).

## 2. Results

### 2.1. Genetic Parameters

The environmental variation coefficient (CVe) showed values below 10% for the nutrients in flowers, except Ca (Table 1). The nutrients Ca, Fe, and Mn had higher values, as follow: 20% for flowers, 33% for grains, 37% for straw, and 43% for leaves. All the other nutrients had values below or close to 20%.

High values of genetic variation coefficient (CVg) were observed for Cu in flowers, Zn in grains, and Mn in straw (Table 1). For leaves, the CVg values were below 20% for all the macronutrients and micronutrients. Lower CVg values were observed for K in flowers and grains, N in straws, and Zn in leaves (Table 1).

Heritability (H^2^) presented the lowest values for the concentration of Cu in flowers, K in grains, N in straw, and Zn in leaves (Table 1). The highest values of H^2^ were observed for the concentration of Mg in flowers, B in grains, P in straw, and Ca in leaves (Table 1).

### 2.2. Leaf Nutrient Concentration over Four Seasons

By analysis of variance, there was a significant interaction between genotypes and leaf collection seasons for all the evaluated nutrients, except for Cu (Table 2). The significant results for the genotypes indicate the existence of variability for the leaf nutrient concentration. This enables studies related to genetic diversity, based on plant nutrition, to be performed.

A relationship was found between the different concentrations of nutrients in coffee leaves as a function of the four sampling times. Higher levels were observed in the first two seasons for the macronutrients N, P, and K, except N, which only had higher averages in season 1. Conversely, the macronutrients Ca, Mg, and S had higher average concentrations in seasons 3 and 4, except for Ca, which was only higher for season 4 (Table 3).

Some genotypes were in the groups with the highest average concentration of at least one macronutrient in all four seasons, as follows: Verdin TA for N, Verdin R for N and P, Clementino for P and Mg, and P1 for K and S (Table 3). Other genotypes had higher average macronutrient concentrations in three of the four seasons, as follows: Pirata for N, P, K, and S, Verdin R and Bamburral for K, Clementino for K and Ca, and Beira Rio 8 for Ca and Mg (Table 3).

In general, N was more concentrated in season 1, decreased in season 2, showed some stability in season 3, and was less concentrated in season 4. On the other hand, A1 and P1 were the only genotypes that showed an increase in season 3 (Table 3).

A slight variation was observed in the concentration of P in leaves over the four seasons, with some genotypes showing a more expressive reduction in season 4. Only Pirata showed a reduction in seasons 3 and 4, Clementino showed a reduction in season 4, and Beira Rio 8 showed a reduction in seasons 2 and 4 (Table 3).

In general, for leaf K concentration, there was a reduction in some genotypes in season 3 and a marked reduction in season 4 in all the genotypes evaluated, except for P1, which was constant over the four seasons (Table 3). All the genotypes showed constant K absorption in the first two seasons. In season 3, only Pirata, Beira Rio 8, Verdin TA, and NV2 showed reductions. Of these, Pirata and Beira Rio reduced in season 4, with 12.09 g kg^−1^ and 8.75 g kg^−1^, respectively (Table 3).

Regarding leaf Ca concentration, the first three seasons presented the lowest absorption along the evaluated times. The second season presented the lowest average for most of the evaluated genotypes (Table 3). Bamburral, Beira Rio 8, Verdin TA, and NV2 showed increased Ca concentration in season 3 (Table 3). The same situation occurred with Verdin R; this genotype already had a low average concentration of Ca in season 1, which reduced even more in season 2, increased in season 3, and increased approximately 80% in the last season, with 23.61 g kg^−1^ (Table 3).

The macronutrient Mg had similar behavior to Ca. Clementino and Beira Rio 8 were the highest means among the genotypes evaluated in season 1, with 2.98 g kg^−1^ and 3.54 g kg^−1^, respectively, with a significant increase in later seasons (Table 3). Additionally, Bamburral, A1, P1, Verdin TA, and NV2 showed increases in seasons 3 and 4, with an average of 6.30 g kg^−1^ and 4.63 g.kg^−1^, respectively. While season 4 is marked by an increase in all the genotypes, Pirata, A1, Clementino, and Beira Rio 8 had higher means around 5.00 g kg^−1^, and the others did not exceed 4.00 g kg^−1^ (Table 3).

For S concentrations, there was generally stability in season 2, followed by a substantial increase in season 3, and reduction in some genotypes and stability of others in season 4 (Table 3). Verdin R and NV2 showed consistently high means in all the evaluated seasons, but had lower means in season 3 among all the genotypes. All the genotypes showed an increase in season 3, and some showed an increase in season 4. Clementino, Beira Rio 8, P1, and Verdin TA showed low means in season 4, as shown in Table 3. The genotype P1 had low averages in season 1, increased slightly in season 2, had a higher average among the genotypes in season 3, with 2.80 g kg^−1^, and a reduction in the last season, with 2.00 g kg^−1^.

Leaf Fe concentration varied according to the genotypes, generally showing the highest mean in seasons 1 and 4. Exceptions were observed for Bamburral, Clementino, and Beira Rio 8, which maintained their concentrations until season 3, increasing in season 4. In all the seasons, the Pirata genotype remained in the group with the highest values, with mean concentrations almost 50% higher than Clementino and P1, which are examples of genotypes with the lowest means in seasons 1, 2 and 4 (Table 4). P1 and NV2 were those that showed a constant mean concentration of Fe over the four seasons.

The genotypes had similar values of Zn in the leaves for seasons 1 and 2, with the exception of the Pirata and Verdin TA genotypes, which presented statistically different averages for these two seasons (Table 4). The genotypes Clementino, Beira Rio 8, and NV2 maintained statistically equal Zn concentrations for all the seasons evaluated (Table 4). Comparing the genotypes in each season, it can be observed that two groups were formed for seasons 1, 2 and 3. The concentration of Zn in the leaves did not differ between the genotypes for season 4. The Clementino genotype stood out for being in the group with the highest averages in all the seasons (Table 4).

Regarding the leaf Cu concentration (Table 4), there was no significant difference between the genotypes in each season. Differences were observed in the seasons when all the genotypes showed the following similar behavior: lower concentration in season 1 (≈7 mg kg^−1^), a slight increase in season 2 (≈10 mg kg^−1^), higher mean in season 3 (≈15 mg kg^−1^), and reduction in season 4 (≈12 mg kg^−1^) (Table 4).

In general, for leaf Mn concentrations, the highest means are observed in season 3, not showing many differences in the other seasons (Table 4). No significant differences were found between the genotypes in season 1. P1 was the genotype that featured in the group with the highest means in all the seasons evaluated. Further, Pirata was in this group, except for season 3, when it was in the group with the lowest average (32.33 mg kg^−1^) (Table 4). Verdin R and Verdin had the lowest means in seasons 2, 3, and 4. All the genotypes, except Pirata, showed an increase in season 3 (Table 4). Regarding leaf B concentration, the highest means of the genotypes were observed in season 2, except for Verdin R, and in season 3, except for Clementino and Beira Rio 8. Verdin R was the only genotype that appeared in the group with the highest mean in all the seasons evaluated, not showing stability in season 2, and showing an increase in season 3, with the highest average among the genotypes (170.33 mg kg^−1^) and a drastic reduction in the last season, with a mean of around 90 mg.kg^−1^. However, Beira Rio 8 featured in the group with the lowest means (Table 4). Only Pirata showed stability over the evaluation seasons.

### 2.3. Nutrient Concentration in Flowers, Grains, and Straw in the Nine Genotypes

According to the analysis of variance, there was a significant difference between the genotypes for the concentration of Ca, Mg, Fe, Zn, Cu, Mn, and B in the flowers (Table 5). For the concentration in the grains, there was a significant difference between the genotypes for the nutrients N, P, Mg, S, Zn, Cu, Mn, and B. Moreover, for the concentration of nutrients in the straw, there was a significant difference for P, K, Ca, Mg, S, Zn, Cu, Mn, and B.

Figure 1 shows the nutrient concentrations in flowers. By clustering the means using the Skott–Knott test, there was no significant difference between the genotypes for the macronutrients N, P, K, and S. For Ca and Mg, the genotypes were grouped into two groups. Conversely, for the micronutrients Zn, Mn, and B, the genotypes were grouped into two groups, and for Fe and Cu, three groups (Figure 1A,B).

The plant tissue of the flowers was noted to concentrate around 38 g kg^−1^ of N, 28 g kg^−1^ of K, and less than 5 g kg^−1^ of the other macronutrients. Regarding the concentration of micronutrients in flowers (Figure 1B), Fe was found at about 100 mg kg^−1^, required approximately 2.5 times more than B, and about five times more than the other micronutrients.

The genotype Verdin R had the highest concentration means of all the nutrients in flowers (Figure 1A). Similarly, Pirata also had high means for all the nutrients evaluated, except for Ca concentration (Figure 1A). On the other hand, the lowest means were found in the concentrations of Ca in Pirata, Bamburral, A1, Clementino, Beira Rio 8, and P1. For Mg, the lowest means were found for Bamburral, A1, P1, Verdin TA, and NV2 (Figure 1A).

The genotypes P1 and Verdin TA had lower means in all the micronutrient concentrations, except Cu for P1 and B for Verdin TA. Regarding the Fe concentrations in flowers, the highest means were observed for Pirata, Verdin R, and A1, while Clementino, P1, Verdin TA, and NV2 had about 25% lower means (Figure 1A).

For the concentration of nutrients in grains, the values were, on average, about 22 g kg^−1^ of N, 14 g kg^−1^ of K, 5 g kg^−1^ of Ca, and less than 3 g kg^−1^ for the other macronutrients (Figure 2A). Fe is found in more significant amounts, six times more than B, regarding the concentration of micronutrients. Zinc is the micronutrient found in lower concentrations in grains (Figure 2A).

Verdin R, Clementino, and Verdin TA had the highest mean N concentrations in grain, while Pirata was isolated from the others with the lowest mean. Regarding P concentration in grains, the group formed by Pirata, Verdin R, A1, and Clementino had the highest mean among the genotypes. K and Ca were the macronutrients with no significant differences in concentrations between genotypes (Figure 2A). In general, Clementino had the highest average concentration in most macronutrients in grains.

Regarding the concentration of micronutrients in grains (Figure 2B), the highest mean for B was observed for the genotypes NV2 and Beira Rio 8, while Pirata had the lowest mean. For Cu, the highest mean was observed for Verdin R and the lowest for the group formed by Bamburral, A1, and Beira Rio 8. No significant differences were observed in the Fe concentration between the genotypes (Figure 2B).

Unlike the other plant tissues presented, the highest macronutrient concentration for straw was K, followed by N and Ca. The other macronutrients presented concentrations below 3 g kg^−1^ (Figure 3A). As for the concentrations of micronutrients in straw, a lower concentration of Fe can be observed compared to other plant organs (Figure 3B). In straw, the average concentration of Fe was around 35 mg kg^−1^, B was around 30 mg kg^−1^, Cu was around 12 mg kg^−1^, Mn was around 8 mg kg^−1^, and Zn was below 2 mg kg^−1^ (Figure 3B).

For macronutrient concentrations in straw, there were two groups for Ca and K, three for S, four for Mg, five for P, and no significant difference between the genotypes for N (Figure 3A). Pirata, Bamburral, Verdin TA, and NV2 had the highest averages for Ca, K, and S, which is a result that they all had in common. The Clementino genotype stands out for presenting the highest average among the genotypes for the concentration of P and Mg in straw.

Although it varied widely, no significant differences were observed between genotypes for Fe concentration in straw (Figure 3B). Two groups were formed for B, Cu, and Zn, and four groups for Mn, emphasizing the Clementino genotype, which participates in the group with the highest averages for all these micronutrients.

### 2.4. Dissimilarity between Genotypes and Relative Contribution of Nutrient Concentration

The clustering of genotypes by the UPGMA hierarchical method, using the Euclidean distance as a measure of dissimilarity, considering the nutrient concentration in flowers, grains, straws, and leaves, allowed the formation of the dendrogram. By establishing a maximum limit of 80% of dissimilarity between the genotypes, the formation of four groups was observed (Figure 4).

The genotypes A1 and Beira Rio 8 formed the first group; Bamburral, Verdin TA, NV2, and P1 formed the second; Clementino was the third; Pirata and Verdin R formed the fourth. The formation of groups, considered divergent between themselves, indicates the variability between the genotypes of *C. canephora* for the concentration of nutrients evaluated.

According to Singh’s [20] method for predicting genetic diversity, the leaf was the plant tissue in which nutrient concentrations contributed most to the diversity among the genotypes (Table 6). Only three variables accounted for 63.3% of the observed variability, presenting the following micronutrients: Fe (31.1%), B (25.7%), and Mn (6.5%). The concentration of Fe in flowers also contributed significantly to the variability (15.2%). The other contributions were below 9%, highlighting the concentration of B in grain (8.2%), the only more expressive concentration for grains. B was the nutrient that contributed most to the variability (38.8%), appearing in the four plant tissues among the eight main contributions.

Among the nutrient correlations in plant tissues, 33% were considered significant (*p* > 0.05), with the majority being positive correlations and classified as moderate (Table 7). P was the macronutrient with the most moderate correlations, which involved its concentration in the flower, grain, and straw, as well as its concentration in the leaf, in the different seasons evaluated.

Of the micronutrients, B had the most significant correlations with the variables flower × straw, leaf—general, 1st, and 2nd seasons; straw × leaf—1st season; leaf—1st season × leaf—2nd and 3rd seasons; leaf—3rd season × 2nd and 4th seasons; moderate negative correlations for grain × leaf—3rd and 4th seasons (Table 7).

Strong correlations were observed for Fe and Mn with the variable flower × leaf—1st season. Mn also showed solid correlations for grain × straw; grain × leaf—general, 2nd, and 4th seasons; straw × leaf—2nd season; leaf—2nd season × leaf—1st and 4th seasons, being the nutrient most correlated with the variables (Table 7). For Mg, strong correlations were observed for the leaf variables (Table 7). The variables leaf 1st season × leaf 2nd season presented most of the solid correlations for N, P, K, Mg, and Mn. Straw × leaf—general showed a strong correlation with K, while straw × leaf—1st season showed a strong correlation with K and Cu.

## 3. Discussion

An occasional increase in nutrient concentrations may have been favored by the increase in precipitation and temperature in the experiment region after September (Figure 5). Coffee plants can have their metabolism reduced in the dry period, due to mild temperatures. At the same time, growth rates are faster in the rainy season and with higher temperatures, due to increased metabolism, as noted by Amaral et al. [21].

Genetic parameters, such as CVe, CVg, and H^2^, are essential tools used in improvement programs, with which possible promising genotypes are identified, contributing to the selection process in genetic improvement [22]. CVe, for example, aims to estimate how much the environment influences the expression of the evaluated characteristics of a given experiment [23]. According to studies by Ferreira et al. [24], CVg is a parameter that can allow inferences about the variability present in the population, and, thus, establishes levels of variability in specific genotypes. The higher the value, the more heterogeneous the evaluated genotypes [22]. The highest CVg values in this study were observed for straw, followed by grains and flowers in some nutrients. However, for most nutrients, the variability was at levels considered low for genetic improvement research [25].

A large part of the nutrients showed high H^2^ values, above 80%. High heritability is associated with how much the phenotypic values represent the genotypic values of individuals [26]. Heritability is one of the most important genetic parameters to verify the genetic control of traits and the efficiency of the selection process in plant improvement. In contrast to most results, the concentration of nutrients in the flowers showed a more significant influence from the environment, a result represented by the lower values of H^2^ for the concentration of many nutrients in this plant organ.

*C. canephora* plants that are highly productive also have greater nutritional demands. These demands vary according to genetic characteristics and plant tissue [4,27]. Further, the nutrient concentration varies according to the edaphic conditions, season, plant age, and genotype maturation cycle [4,10], indicating that some genotypes can absorb or translocate more nutrients than others.

Considering all tissues, the orders of concentration of the macronutrients and micronutrients found in this work were N > K > Ca > P = Mg = S and Fe > B > Mn > Cu > Zn, respectively. N was the most required nutrient during the evaluation period, around 40 g kg^−1^ in flowers, and 20 g kg^−1^ in leaves and grains. The exception was straw, in which the most commonly found nutrient was K (about 12 g kg^−1^).

In flowers, N and K were found in higher concentrations (about 35 and 30 g kg^−^¹, respectively) than the other macronutrients that, together, concentrated less than 20 g kg^−^¹. While P is not the most required nutrient during the production cycle, it plays a vital role in this phase, and plays essential roles in plants, acting in the functions of energy metabolism in the form of ATP, which is essential in the growth of plant organs [16].

In the present study, no differences were observed between genotypes for the N, K, P, and S concentration in flowers. Supporting the results, Laviola et al. [28] also observed similar values for these macronutrients among cultivars of *C. arabica*. As for the micronutrients, mainly Fe, differences between genotypes are observed; four genotypes (Clementino, P1, Verdin TA, and NV2) had about 20% less Fe than the genotypes Pirata, Verdin R, and A1.

The development of flower buds occurs concomitantly with the vegetative growth of plants. They can compete for water, nutrients and carbohydrates, and can be used as indicators for nutritional diagnosis, as they are a substantial drain within the plant [28]. There are nutrients, such as Ca and B, that follow the transpiration flow; however, concerning the functionality of the conducting vessels, they are much smaller in flowers than in leaves, making transport difficult. Thus, during the formation of flower buds, it is interesting that the plant has already received and absorbed the necessary amount of nutrients, since the absorption of nutrients starts before floral anthesis. *C. canephora* genotypes have high genetic variability due to factors such as allogamy, which makes nutrient absorption, or translocation, and flowering differentiated [2,19].

The highest nutrient values found in concentrations of flowers, grains, straw, and leaves can determine favorable conditions for identifying the diversity within these genotypes [29]. The present study can be compared to that by Covre et al. [15], who evaluated *C. canephora* grains and straws, obtaining the macronutrients K, N, and Ca as the highest in straws, and N and K in grains. For micronutrients, the authors also found Fe and B to be the most accumulated in straws and grains. These higher levels of nutrients are believed to be due to the fertilization in previous months, contributing to seed filling. Moreover, in this production stage, fertilization has already been carried out according to the fertilization recommendation for the specific crop [30].

The more significant amount of N in coffee grains than in straw, and the more significant amount of K in straw, may be due to several factors, such as being linked to the requirement of the plant itself for higher concentrations of N for production [31]. The high concentration of K in straw is believed to be due to the pre-existing high content in leaves and fruits [32]. Thus, coffee straw can be used in crops as soil organic matter, due to its high concentrations of K and N.

The micronutrient Fe had the highest concentration in *C. canephora* grains and straw, which confirms the studies by Covre et al. [15] and Partelli et al. [5], which found that Fe is the most accumulated micronutrient in conilon coffee. When evaluating genotypes with different maturation cycles, Marré et al. [3] also found a higher Fe concentration in *C. canephora* genotypes. B was the second most accumulated, followed by Mn. These results are similar to those found by Covre et al. [15] in the grains and straws of irrigated and non-irrigated coffee trees.

After flowering, and until grain maturation in coffee genotypes, a reduction in N and K concentrations in leaves is noted, reflecting the translocation of the photoassimilates needed to ensure fruit quality formation [10].

Leaf N concentration was relatively within what is expected for the crop in the flowering and grain filling stages [8]. According to Martins et al. [7], there is a direct relationship between the absorption of nutrients and the plant cycle, considering that the maturation of the fruits requires more significant amounts of metabolites, which intensifies the transport of photoassimilates to the fruits.

Oscillations in the concentrations of N and K in the leaves were also observed between the genotypes Pirata and Bamburral, which are known to have late and medium/late maturation, with a decrease in leaf concentrations later compared to the other genotypes. These fluctuations can be attributed to the maturation conditions of the grains, conditioned to the genetic characteristics of each genotype. According to Partelli et al. [4,5], the highest nutrient absorption speed was verified in fruits with early maturation and higher growth rates. Genotypes may present nutritional variations throughout the year, with influence from the type of maturation (early, late, and medium) [4,10], climatic factors, or the phenological phase itself. Thus, different genotypes need nutrient concentrations at different periods, but the period is almost always similar to their phenological phase [4,30]. Oliosi et al. [10] studied the seasonal variation relating to leaf concentrations, obtaining higher N concentrations in early and medium maturation genotypes in the first samplings, and lower in late-maturing genotypes, with a higher peak in November for the latter.

In leaves, the macronutrient P is required more in the periods before flowering, in June, July, and the grain filling stages, with a higher concentration at the end of January [18], which is similar to the studies of the present work. In season 3 (February), there was an increase in the concentration of this nutrient. While P is the macronutrient required in the smallest amount [14], it plays a vital role in photosynthesis and protein phosphorylation [16].

As for the concentrations of Ca and Mg present in the leaves, both showed similarities in the seasons evaluated [14]. Ca showed a reduction in the fruit set phases for leaf contents and Mg, which, although uncommon, may reflect the redistribution of nutrients to the fruits [9]. Oliosi et al. [10] stated, in their studies, that the highest concentrations of Ca were in October and November, and Mg decreased in September, followed by an increase.

The S contents in the leaves are found to have higher concentrations for grain filling and seed filling periods, as in the studies by Oliosi et al. [10] and Silva et al. [25]. The concentration of this macronutrient showed variations due to the maturation of the genotypes. In early, medium, late, and super late cycle conilon coffee, Partelli et al. [4] and Oliosi et al. [10] also obtained this variation, with higher concentrations of S in the leaf for the genotypes V8 and V12, which are considered to have an early maturation cycle. According to Tomaz et al. [33], different genotypes may differ concerning the absorption, translocation, and efficiency of S. This effect is due to affinities in absorption by the root system, mobility in the xylem, and phloem vessels [34].

Thus, the seasonality observed in the concentration of macronutrients in leaves varies according to the seasons and genotypes evaluated in this study. The hottest periods of the year and early spring coincide with the highest concentrations of the macronutrients N, P, and K. After September, there is an increase in precipitation (Figure 5), favoring greater nutrient absorption, which justifies the higher concentrations found for the macronutrients Ca, Mg, and S. Furthermore, in this season, according to Marenco and Lopes [35], the increase in temperature causes acceleration of the respiratory activities of plants and better metabolism of the root system.

The micronutrient Fe is considered to be the most accumulated by conilon coffee plants. According to Bragança et al. [36], approximately 80% of Fe accumulates in chloroplasts, which have a vital role in photosynthesis and chlorophyll production. According to Covre et al. [9], low micronutrient levels in coffee are observed at the end of the crop cycle. In this study, this situation occurs in season 4, the fruit maturation period. According to Oliosi et al. [10], the Fe concentrations in leaves showed higher concentrations soon after it rained.

In studies with coffee trees in the Western Amazon, Dubberstein et al. [37] noted that Cu showed slight variation and linear behavior, remaining constant since the first evaluation. This characteristic situation of Cu also occurred in the present study.

In studies with two evaluation periods, Marré et al. [3] observed that the concentrations of Mn in the leaves were reduced, which could be due to the greater demand of the micronutrient in the development of the fruits during this period, the maturation period. Oliosi et al. [10] observed a reduction in B in August, mainly due to flower formation.

In most micronutrients (Fe, Zn, B, and Mn), lower levels in the first seasons (fruit set phase) and a high range could be verified, characterized as expansion and seed filling of the fruits. This behavior is characteristic of conilon coffee trees in Brazil, as already observed in studies in the northern region of Espírito Santo [3] and the Western Amazon [37].

The formation of groups by the dendrogram indicates variability among the *C. canephora* genotypes for nutritional contents of leaves, flowers, grains, and straw. Silva et al. [25], Gomes et al. [8], and Martins et al. [2] also observed divergence between the conilon coffee genotypes for nutrient concentration. These studies are essential for improvement programs, being used to select genotypes with superior characteristics.

Of the four groups formed in the present study, the one composed of Pirata and Verdin R presented the highest concentrations of nutrients in the evaluated tissues, and the slightest oscillation in the nutritional dynamics in their leaves. Pirata and Verdin R have similar maturation cycles, early and medium/early, respectively. Two other genotypes, which form another group, also share the following similar maturation cycles: medium cycle, A1; medium/early cycle, Beira Rio 8.

Several correlations were observed for the concentration of nutrients between the evaluated plant organs. Most of the correlations were positive and involved the concentration of nutrients in the leaves and other plant organs. The concentration of nutrients in the leaves is commonly used for the nutritional diagnosis of coffee plants [8]. Thus, an adequate concentration of nutrients in the coffee leaves must be maintained, especially during periods of greater nutritional demand, such as flowering and grain filling [19,28].

While this study has addressed the nutritional similarities in different organs of the plant, and has, thus, obtained dissimilarity-based groups, factors such as leaf morphology, and roots and plant architecture were not addressed. These morphological attributes act directly on nutrient absorption, photosynthetic rates, and radiation resistance, and can affect the nutritional status of each plant’s tissues [38]. Thus, deepening the morphological parameters may bring greater understanding to the nutritional discussion of conilon coffee genotypes.

## 4. Materials and Methods

### 4.1. Study Area and Experiment Description

The experiment was conducted in São Mateus, Espírito Santo, latitude 18°40′25″ S, longitude 40°51′23″ W, and an average altitude of approximately 36 m. The local soil is classified as Yellow Argisol with a sandy loam texture, with undulating relief [39], whose chemical and physical attributes extracted from the 0–10 cm soil layer are presented in Table 8. According to the Koppen classification, the region’s climate is hot and humid, Aw type, with a dry season in autumn–winter and a rainy season in spring–summer [40].

Figure 5 shows the average values of minimum, mean, maximum temperature, radiation, precipitation, evapotranspiration, and relative air humidity collected in the meteorological station located in the experimental farm.

In June 2018, planting an experimental crop consisting of *C. canephora* genotypes, cultivated under full sun conditions, was carried out with the spacing of two meters between rows and one meter between plants (2 × 1 m). The fertilization of the experimental area followed the technical recommendations suggested for the culture [41]. The other cultural practices consisted of weed control with herbicides and mowing, preventive phytosanitary management, liming, and drip irrigation.

### 4.2. Sample Design

The experiment was conducted in a randomized block design with three replications and treatments consisting of nine genotypes. Each experimental plot consisted of three plants. The following genotypes evaluated in this study belong to two conilon coffee cultivars: Bamburral, Beira Rio 8, Clementino, Pirata, and Verdim R belong to the Tributun cultivar [1,13], NV2, P1, and Verdim TA belong to the cultivar called Andina [42,43], and A1 participates in the composition of both cultivars.

### 4.3. Evaluations of Plant Organs

Samples of leaves, flowers, and fruits were collected to determine the concentration of nutrients in plant tissues. Leaf samples were collected at three-month intervals starting in August 2019 (season 1), November 2019 (season 2), February 2020 (season 3), and May 2020 (season 4). They were collected on both sides of the planting line, the middle third of the plants, 3rd and 4th pair of leaves from the apex of the plagiotropic branches, totaling four leaves on each side of the plant. Flower samples were collected in July 2019, right after floral anthesis. The fruits were collected in the middle third of the plant in the month of June 2020, when they presented complete maturation for all genotypes.

The materials were packed in paper bags correctly identified and dried in an oven with forced air circulation at 65 °C until a constant mass was reached. The fruits were manually peeled, separating the grains and straw. The plant materials were sent to determine the concentration of nitrogen (N), phosphorus (P), potassium (K), calcium (Ca), magnesium (Mg), sulfur (S), iron (Fe), zinc (Zn), copper (Cu), manganese (Mn), and boron (B), according to the methodology described by Silva [44]. The N content was determined after sulfuric acid digestion using the Nessler colorimetric method; P by molecular absorption spectrophotometry; K by flame photometry; S by sulfate turbidimetry. The contents of Ca, Mg, Fe, Zn, Mn, and Cu were determined by atomic absorption spectrophotometry, and B by colorimetry using the Azometin-H method.

### 4.4. Statistical Analyses

The values of concentrations of grains, straw, flowers, and leaves were subjected to analysis of variance by the F test (*p* ≤ 0.01) to identify differences between genotypes. The Scott–Knott algorithm (*p* ≤ 0.05) clustered the nutrient concentration means among the genotypes. From the variance components, the following parameters were estimated for each characteristic: the experimental variation coefficient (CVe), which estimates the influence of the environment; the genetic variation coefficient (CVg), which estimates the genetic variability; heritability (H^2^), which estimates the genetic control of the evaluated characteristics. The parameters were estimated according to the following formulas:

(a)CVe (%) = (σ_e_/M) × 100where σ_e_ = standard deviation of the experimental residue; M = experimental average.(b)CVg (%) = (σ_g_/M) × 100where σ_g_ = genetic standard deviation; M = experimental average.(c)H^2^ (%) = (σ^2^_g_/σ^2^_F_) × 100where σ^2^_g_ = genotypic variance component; σ^2^_F_ = component of phenotypic variance.

For the study of genetic diversity, the Euclidean distance matrix was established as a dissimilarity measure, and the genotypes were clustered using the unweighted pair group method using arithmetic averages (UPGMA) hierarchical method. The study of the relative importance of nutrient concentrations was applied to predict genetic diversity, according to Singh [20]. Spearman’s non-parametric correlation analysis was used to assess the degree to which nutrient concentrations in different plant organs were correlated with each other. All statistical analyses were performed using the Genes computer program [45].

## 5. Conclusions

In general, the order of concentration of the macronutrients and micronutrients was N > K > Ca > P = Mg = S and Fe > B > Mn > Cu > Zn, respectively, showing oscillations as a function of each genotype.

N, followed by K, were the nutrients found in higher concentrations in flowers, grains, and leaves, with K being the first in fruit straw. Fe was the micronutrient found in the highest concentrations in the organs studied.

In leaves, in general, the concentrations of nutrients were high in the first seasons, decreasing in the last ones. Furthermore, fluctuations in these leaf concentrations were observed to be due to some genotypes’ early or late maturation cycles.

The concentration of micronutrients in leaves accounted for more than 75% of all the genetic diversity among the genotype groups.

For a nutritional diagnosis, it is important to take comparisons of the genetic diversity and evaluation periods into consideration.

## Figures and Tables

**Figure 1 plants-10-02661-f001:**
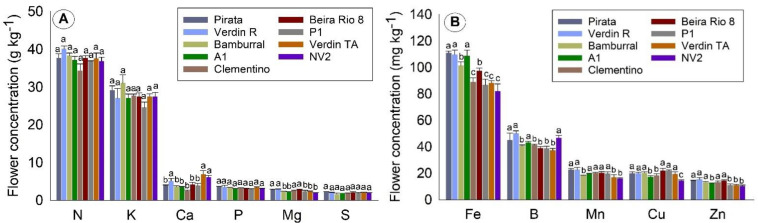
Concentration of macronutrients (**A**) and micronutrients (**B**) in flowers of nine *Coffea canephora* genotypes. São Mateus, state of Espírito Santo—Brazil. Means followed by the same letter for each nutrient do not differ by the Skott–Knot test at 5% probability.

**Figure 2 plants-10-02661-f002:**
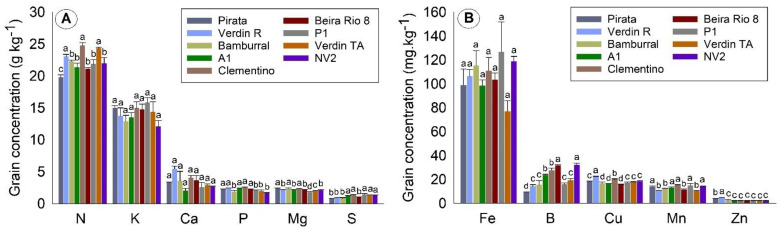
Concentration of macronutrients (**A**) and micronutrients (**B**) in grains of nine *Coffea canephora* genotypes. São Mateus, state of Espírito Santo—Brazil. Means followed by the same letter for each nutrient do not differ by the Skott–Knot test at 5% probability.

**Figure 3 plants-10-02661-f003:**
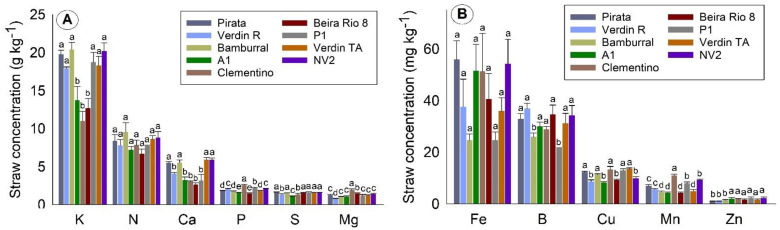
Concentration of macronutrients (**A**) and micronutrients (**B**) in the straw of nine *Coffea canephora* genotypes. São Mateus, state of Espírito Santo—Brazil. Means followed by the same letter for each nutrient do not differ by the Skott–Knot test at 5% probability.

**Figure 4 plants-10-02661-f004:**
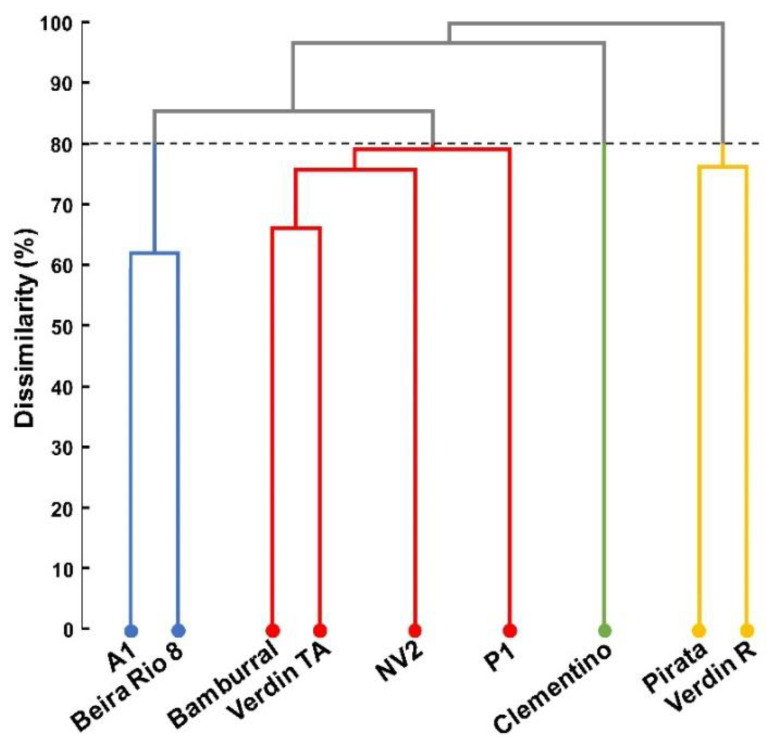
Dendrogram representing the dissimilarity between 9 genotypes of *Coffea canephora*, obtained by the UPGMA clustering method with Euclidean distance, considering the concentration of nutrients in flowers, grains, straws, and leaves.

**Figure 5 plants-10-02661-f005:**
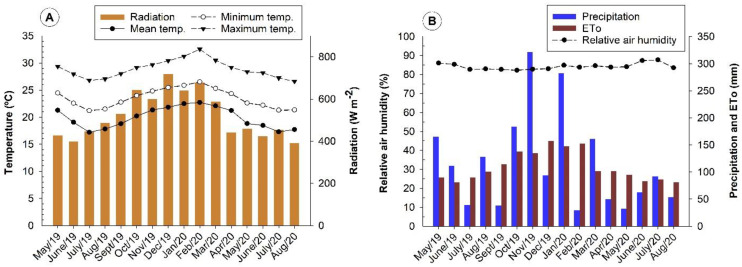
Solar radiation, mean, maximum, and minimum temperature values (**A**), precipitation, evapotranspiration (ETo), and relative humidity (**B**) occurred during the experiment period (2019/2020). São Mateus—ES. Drip irrigation was used to ensure the maintenance of an adequate level of soil moisture. The water balance was calculated daily based on crop evapotranspiration, precipitation, and soil water storage characteristics.

**Table 1 plants-10-02661-t001:** Estimates of environmental variation coefficient (CVe), genetic variation coefficient (CVg), and genotypic determination coefficient (H^2^) for nutrient concentration in flowers, grains, straw, and leaves in nine *Coffea canephora* genotypes. São Mateus, state of Espírito Santo—Brazil.

Nutrients	Flowers			Grains			Straw			Leaves		
	CVe	CVg	H^2^	CVe	CVg	H^2^	CVe	CVg	H^2^	CVe	CVg	H^2^
	%											
N	4.56	3.06	57.50	3.80	6.68	90.27	15.51	6.08	31.51	9.45	9.16	91.84
P	8.79	5.84	56.97	6.63	12.22	91.07	3.53	16.50	98.50	12.07	9.00	86.97
K	9.45	3.04	23.70	12.48	4.15	24.94	12.23	19.57	88.48	10.58	10.05	91.55
Ca	20.10	28.71	85.96	33.19	22.95	58.92	14.16	30.43	93.27	15.11	15.48	92.65
Mg	8.06	12.10	87.13	4.87	7.54	87.76	11.92	23.29	91.97	20.61	15.94	87.77
S	9.99	6.92	59.02	12.72	19.81	87.90	8.38	11.92	85.86	12.86	9.31	86.29
Fe	5.21	11.01	81.71	18.63	8.24	36.98	37.01	19.70	45.96	20.50	13.99	84.82
Zn	9.41	12.52	84.16	14.64	41.40	96.00	29.07	24.94	68.82	14.55	6.01	67.20
Cu	8.74	83.56	11.37	5.46	10.84	92.21	8.94	18.01	92.41	21.38	9.02	68.11
Mn	9.73	10.11	76.39	8.01	13.39	89.34	12.47	35.81	96.12	43.52	19.19	69.99
B	9.11	8.29	71.31	13.00	37.83	96.22	12.33	13.42	78.06	15.37	14.01	90.88

**Table 2 plants-10-02661-t002:** Summary of analysis of variance for leaf nutrient concentration in nine *Coffea canephora* genotypes for four seasons. São Mateus, state of Espírito Santo—Brazil.

Nutrients	Mean Square					Mean
	Genotypes (G)	Seansons (Se)	G × Se	Error (G)	Error (Se)	
N	51.40 **	305.99 **	9.58 **	4.19	3.13	21.66
P	0.26 **	0.09 **	0.04 *	0.03	0.02	1.54
K	24.35 **	124.85 **	3.43 **	2.06	1.36	13.56
Ca	90.26 **	1707.46 **	30.55 **	6.64	5.91	17.05
Mg	4.64 **	30.36 **	0.46 **	0.57	0.19	3.65
S	0.34 **	2.57 **	0.16 **	0.05	0.05	1.68
Fe	3017.45 **	9453.00 **	1187.19 **	458.09	350.56	104.40
Zn	1.29 *	5.22 **	1.76 **	0.42	0.50	4.47
Cu	17.11 ^ns^	257.88 **	7.73 ^ns^	5.46	5.58	10.93
Mn	625.08 *	2733.64 **	231.14 **	187.58	58.96	31.47
B	2491.23 **	7539.62 **	731.37 **	227.20	153.33	98.06

^ns^, ** and *, not significant, significant at 1 and 5% probability, respectively, by the F test.

**Table 3 plants-10-02661-t003:** Leaf concentration of N, P, K, Ca, Mg, and S (g kg^−1^) in nine *Coffea canephora* genotypes for four seasons of the year. São Mateus, state of Espírito Santo—Brazil.

**Genotypes**	**Nutrients**											
	**N**				**P**				**K**			
	**August 2019**	**November 2019**	**February 2020**	**May 2020**	**August 2019**	**November 2019**	**February 2020**	**May 2020**	**August 2019**	**November 2019**	**February 2020**	**May 2020**
Pirata	27.28 Aa	26.58 Aa	21.58 Bb	19.04 Ab	1.81 Aa	1.87 Aa	1.48 Bb	1.60 Ab	17.71 Aa	16.67 Aa	14.79 Ab	12.09 Bc
Verdin R	27.49 Aa	23.61 Ab	25.01 Ab	19.44 Ac	1.90 Aa	1.70 Aa	1.59 Aa	1.72 Aa	16.04 Ba	16.04 Aa	14.17 Aa	11.25 Bb
Bamburral	19.93 Ca	18.88 Ba	20.91 Ba	15.26 Bb	1.49 Ba	1.46 Ba	1.36 Ba	1.30 Ba	16.88 Aa	15.83 Aa	15.63 Aa	9.79 Cb
A1	23.96 Ba	18.78 Bb	22.61 Ba	17.29 Ab	1.49 Ba	1.39 Ba	1.38 Ba	1.45 Aa	14.79 Ba	15.84 Aa	14.80 Aa	9.59 Cb
Clementino	24.06 Ba	17.55 Bb	18.50 Bb	14.21 Bc	1.87 Aa	1.82 Aa	1.80 Aa	1.52 Ab	12.50 Ca	12.71 Ba	11.46 Ba	8.54 Cb
Beira Rio 8	27.21 Aa	20.42 Bc	23.64 Ab	16.47 Bd	1.51 Ba	1.24 Bb	1.47 Ba	1.24 Bb	12.71 Ca	13.96 Ba	11.05 Bb	8.75 Cc
P1	22.47 Bb	19.37 Bc	25.76 Aa	17.73 Ac	1.38 Ba	1.36 Ba	1.61 Aa	1.52 Aa	15.21 Ba	15.42 Aa	13.75 Aa	14.17 Aa
Verdin TA	28.63 Aa	24.83 Ab	25.06 Ab	18.01 Ac	1.55 Ba	1.43 Ba	1.56 Aa	1.51 Aa	15.42 Ba	16.04 Aa	12.50 Bb	12.29 Bb
NV2	27.32 Aa	21.70 Bb	21.07 Bb	18.20 Ac	1.52 Ba	1.41 Ba	1.52 Ba	1.42 Aa	14.80 Ba	15.00 Aa	10.42 Bb	9.59 Cb
Average	25.37	21.3	22.68	17.29	1.61	1.52	1.53	1.48	15.12	15.28	13.17	10.67
**Genotypes**	**Ca**				**Mg**				**S**			
	**August 2019**	**November 2019**	**February 2020**	**May 2020**	**August 2019**	**November 2019**	**February 2020**	**May 2020**	**August 2019**	**November 2019**	**February 2020**	**May 2020**
Pirata	11.96 Ab	9.32 Cb	12.38 Bb	17.64 Da	2.38 Bd	3.21 Cc	3.92 Bb	5.15 Aa	1.68 Ab	1.76Ab	2.14 Ca	2.23 Aa
Verdin R	13.02 Ab	8.34 Cc	14.29 Bb	23.61 Ca	2.25 Bd	3.13 Cc	3.77 Bb	4.50 Ba	1.42 Aa	1.47 Aa	1.55 Da	1.42 Ba
Bamburral	9.76 Bc	6.97 Cc	14.91 Bb	33.96 Aa	1.75 Bb	2.33 Db	3.92 Ba	4.48 Ba	1.30 Ab	1.36 Ab	1.96 Ca	1.92 Aa
A1	13.25 Ab	13.33 Bb	15.35 Bb	33.96 Aa	2.10 Bc	3.52 Cb	5.02 Aa	5.50 Aa	1.26 Ab	1.45 Ab	2.11 Ca	1.91 Aa
Clementino	13.15 Ac	20.38 Ab	20.09 Ab	30.51 Ba	2.98 Ab	5.11 Aa	5.11 Aa	5.06 Aa	1.33 Ab	1.63 Ab	2.39 Ba	1.50 Bb
Beira Rio 8	14.47 Ac	13.42 Bc	21.55 Ab	32.18 Aa	3.54 Ab	4.17 Bb	4.92 Aa	5.42 Aa	1.42 Ab	1.30 Ab	2.10 Ca	1.33 Bb
P1	6.57 Bc	11.65 Bb	14.27 Bb	28.47 Ba	1.77 Bb	2.08 Db	3.59 Ba	3.92 Ba	1.23 Ac	1.68 Ab	2.80 Aa	2.00 Ab
Verdin TA	9.36 Bd	14.00 Bc	19.28 Ab	29.58 Ba	2.52 Bb	3.15 Cb	4.77 Aa	4.29 Ba	1.50 Ab	1.50 Ab	2.07 Ca	1.55 Bb
NV2	8.26 Bc	11.81 Bc	16.70 Bb	26.06 Ca	1.88 Bb	2.52 Db	3.94 Ba	3.84 Ba	1.43 Aa	1.50 Aa	1.80 Da	1.40 Ba
Average	11.09	12.14	16.54	28.44	2.35	3.25	4.33	4.68	1.4	1.52	2.1	1.7

Means followed by equal letters, lowercase in the row, and uppercase in the column for each nutrient, do not differ by Scott–Knott at 5% probability.

**Table 4 plants-10-02661-t004:** Leaf concentration of Fe, Zn, Cu, Mn, and B (mg kg^−1^) in nine *Coffea canephora* genotypes for four seasons. São Mateus, state of Espírito Santo—Brazil.

**Genotypes**	**Nutrients**											
	**Fe**				**Zn**				**Cu**			
	**August 2019**	**November 2019**	**February 2020**	**May 2020**	**August 2019**	**November 2019**	**February 2020**	**May20 20**	**August 2019**	**November 2019**	**February 2020**	**May 2020**
Pirata	13200 Ab	140.67 Ab	100.33 Ac	182.00 Aa	5.33 Ab	4.00 Bc	6.67 Aa	4.00 Ac	7.67 Ad	11.33 Ac	12.67 Aa	11.67 Ab
Verdin R	12900 Aa	66.00 Bb	79.00 Ab	130.33 Ba	4.00 Bb	4.33 Bb	5.67 Aa	4.33 Ab	6.67 Ad	9.33 Ac	14.00 Aa	10.67 Ab
Bamburral	7900 Bb	52.00 Bb	81.67 Ab	148.33 Aa	3.67 Bb	3.33 Bb	6.00 Aa	4.00 Ab	7.67 Ad	10.33 Ac	17.00 Aa	10.67 Ab
A1	12300 Aa	88.33 Bb	87.67 Ab	117.00 Ba	3.67 Bb	3.33 Bb	5.67 Aa	4.67 Aa	6.00 Ad	8.33 Ac	15.00 Aa	14.00 Ab
Clementino	8300 Bb	72.67 Bb	89.67 Ab	124.00 Ba	5.00 Aa	4.67 Aa	5.33 Aa	4.67 Aa	7.00 Ad	8.67 Ac	13.67 Aa	9.33 Ab
Beira Rio 8	10733 Ab	93.00 Bb	112.67 Ab	155.00 Aa	5.00 Aa	4.00 Ba	4.67 Ba	4.67 Aa	6.67 Ad	9.00 Ac	12.67 Aa	9.67 Ab
P1	6500 Ba	80.67 Ba	98.67 Aa	102.67 Ba	3.67 Bb	3.67 Bb	4.67 Ba	5.00 Aa	9.00 Ad	14.33 Ac	16.00 Aa	14.67 Ab
Verdin TA	7733 Bb	111.33 Aa	118.00 Aa	118.00 Ba	4.33 Bb	5.67 Aa	3.67 Bb	3.67 Ab	8.33 Ad	8.67 Ac	16.33 Aa	10.33 Ab
NV2	8467 Ba	109.00 Aa	108.67 Aa	110.67 Ba	4.00 Ba	4.00 Ba	3.67 Ba	4.33 Aa	7.33 Ad	8.00 Ac	16.33 Aa	14.33 Ab
Average	6-Apr-00	30-Mar-00	6-Apr-00	11-May-00	4-Jan-00	4-Jan-00	5-Jan-00	4-Jan-00	7-Jan-00	9-Jan-00	14-Jan-00	11-Jan-00
**Genotypes**	**Mn**				**B**							
	**August 2019**	**November 2019**	**February 2020**	**May 2020**	**August 2019**	**November 2019**	**February 2020**	**May 2020**				
Pirata	3067 Ab	31.00 Ab	32.33 Cb	52.33 Aa	95.33 Ba	108.00 Ba	107.67 Ca	92.00 Aa				
Verdin R	2333 Aa	24.67 Ba	28.00 Ca	16.67 Ca	121.67 Ab	120.00 Ab	170.33 Aa	90.33 Ac				
Bamburral	1633 Ab	17.67 Bb	52.00 Ba	21.67 Cb	63.33 Cb	94.67 Ba	110.67 Ca	93.67 Aa				
A1	1867 Ab	19.67 Bb	32.67 Ca	31.00 Ca	92.67 Bb	137.00 Aa	134.00 Ba	97.00 Ab				
Clementino	2700 Ab	40.67 Aa	48.67 Ba	36.00 Bb	93.67 Bb	128.33 Aa	105.00 Cb	55.67 Bc				
Beira Rio 8	24.33 Ab	20.33 Bb	47.33 Ba	24.33 Cb	104.00 Ba	108.00 Ba	85.00 Db	44.67 Bc				
P1	27.00 Ac	33.67 Ac	69.00 Aa	46.33 Ab	71.33 Cb	91.67 Ba	98.33 Da	79.00 Ab				
Verdin TA	19.00 Ab	24.33 Bb	37.00 Ca	18.67 Cb	85.00 Cb	108.00 Ba	121.33 Ca	88.00 Ab				
NV2	19.00 Ab	25.33 Bb	64.33 Aa	32.00 Cb	74.00 Cb	92.67 Ba	95.67 Da	72.33 Ab				
Average	22.81	26.37	45.70	31.00	89.00	109.82	114.22	79.19				

Means followed by equal letters, lowercase in the row, and uppercase in the column for each nutrient, do not differ by Scott–Knott at 5% probability.

**Table 5 plants-10-02661-t005:** Summary of analysis of variance for nutrient concentration in flowers, grains and straw of nine *Coffea canephora* genotypes. São Mateus, state of Espírito Santo—Brazil.

Nutrients	Flower			Grain			Straw		
	Mean Square		Mean	Mean Square		Mean	Mean Square		Mean
	Genotypes	Error		Genotypes	Error		Genotypes	Error	
N	2.35 ^ns^	2.90	37.4	10.27 **	0.72	22.29	1.46 ^ns^	1.57	8.09
P	2.32 ^ns^	0.08	3.29	11.19 **	0.02	2.19	66.52 **	0.01	1.96
K	1.31 ^ns^	6.85	27.69	1.33 ^ns^	3.12	14.15	8.68 **	4.32	16.99
Ca	7.12 **	0.80	4.44	2.43 ^ns^	1.25	3.37	14.86 **	0.38	4.34
Mg	7.77 **	0.04	2.52	8.17 **	0.01	2.18	12.46 **	0.02	1.29
S	2.44 ^ns^	0.04	2.00	8.27 **	0.02	1.15	7.07 **	0.01	1.48
Fe	14.41 **	25.57	97.11	1.59 ^ns^	391.95	106.26	1.85 ^ns^	240.36	41.89
Zn	6.31 **	1.45	12.81	25.00 **	0.15	2.63	3.21 *	0.24	1.70
Cu	6.08 **	2.84	19.3	12.83 **	0.98	18.15	13.18 **	0.99	11.11
Mn	4.24 **	3.65	19.63	9.38 **	1.08	12.96	25.74 **	0.68	6.59
B	3.49 *	14.94	42.44	26.42 **	7.53	21.11	4.56 **	14.43	30.81

^ns^, ** and *, not significant, significant at 1 and 5% probability, respectively, by the F test.

**Table 6 plants-10-02661-t006:** Relative contribution of nutrient concentration in flowers, grains, straw, and leaves to genetic diversity in 9 genotypes of *Coffea canephora*, according to Singh’s (1981) method, Euclidean distance. São Mateus, state of Espírito Santo—Brazil.

Variables	S.j ¹	Value (%)	Accumulated Value (%)
Leaf Fe	18,104.72	31.13	31.13
Leaf B	14,947.37	25.70	56.83
Flower Fe	884.00	15.20	72.03
Grain B	4774.00	8.21	80.24
Leaf Mn	3750.50	6.45	86.69
Straw B	1578.22	2.71	89.41
Flower B	1250.00	2.15	91.56
Straw K	899.63	1.55	93.10
Other variables	4011.97	6.90	100.00

^1^ S.j (Singh, 1981, [21]).

**Table 7 plants-10-02661-t007:** Spearman correlation coefficients for nutrient concentration in flowers, grains, straw, and leaves of *Coffea canephora* genotypes. São Mateus, state of Espírito Santo—Brazil.

Variables	N	P	K	Ca	Mg	S	Fe	Cu	Mn	Zn	B
Flower × Grain	−0.19	0.09	−0.18	−0.12	0.32	0.04	−0.19	−0.18	0.01	0.56 **	−0.12
Flower × Straw	0.06	−0.14	0.14	0.50 **	0.05	0.09	0.03	0.22	−0.07	−0.36	0.43 *
Flower × Leaf—General	0.38 *	0.41 *	0.14	−0.18	0.44 *	0.23	0.39 *	−0.04	0.15	0.57 **	0.41 *
Flower × Leaf—1st season	0.11	0.45 *	0.44 *	−0.23	0.51 **	0.48 *	0.63 **	0.38 *	0.60 **	0.36	0.40 *
Flower × Leaf—2nd season	0.38 *	0.41 *	0.15	−0.18	0.41 *	0.24	0.09	0.54 **	0.33	−0.15	0.23
Flower × Leaf—3rd season	0.22	0.05	0.13	0.02	0.14	0.07	−0.33	−0.35	−0.40 *	0.63 **	0.36
Flower × Leaf—4th season	0.44 *	0.27	−0.23	−0.18	0.44 *	0.03	0.42 *	−0.29	0.30	0.12	0.15
Grain × Straw	0.21	0.10	−0.18	−0.18	0.18	−0.10	−0.15	0.04	0.69 **	0.62 **	0.22
Grain × Leaf—General	−0.21	0.46 *	0.12	−0.09	0.26	0.07	−0.28	−0.11	0.75 **	0.22	−0.38
Grain × Leaf—1st season	−0.10	0.44 *	−0.06	0.25	0.16	−0.35	−0.17	−0.09	0.38 *	0.04	0.00
Grain × Leaf—2nd season	−0.24	0.36	−0.01	−0.10	0.25	0.21	−0.15	−0.11	0.62 **	−0.03	0.07
Grain × Leaf—3rd season	−0.01	0.31	0.12	0.14	0.12	0.39 *	−0.38 *	0.01	0.51 **	0.41 *	−0.40 *
Grain × Leaf—4th season	−0.30	0.29	0.27	−0.09	0.33	−0.27	−0.10	−0.10	0.63 **	−0.18	−0.59 **
Straw × Leaf—General	0.06	0.58 **	0.63 **	−0.37	0.37	0.25	0.36	0.25	0.57 **	−0.07	0.29
Straw × Leaf—1st season	−0.11	0.40 *	0.75 **	−0.43 *	0.47 *	0.35	0.47 *	0.66 **	0.31	−0.05	0.46 *
Straw × Leaf—2nd season	0.09	0.42 *	0.54 **	−0.32	0.41 *	0.11	0.34	0.35	0.65 **	−0.07	0.30
Straw × Leaf—3rd season	0.04	0.64 **	0.28	−0.30	0.35	0.16	0.02	0.02	0.36	−0.20	0.24
Straw × Leaf—4th season	0.19	0.37	0.42 *	−0.37	0.06	0.09	0.05	−0.07	0.38 *	0.36	−0.13
Leaf—1st season × Leaf—2nd season	0.63 **	0.68 **	0.72 **	0.24	0.76 **	0.06	0.18	0.36	0.73 **	0.21	0.49 **
Leaf—1st season × Leaf—3rd season	0.21	0.37	0.53 **	0.22	0.61 **	−0.37	0.02	−0.10	0.10	0.21	0.49 **
Leaf—1st season × Leaf—4th season	0.40 *	0.44 *	0.39 *	0.03	0.49 **	0.05	0.41 *	0.13	0.55 **	0.08	−0.13
Leaf—2nd season × Leaf—3rd season	0.25	0.34	0.50 **	0.49 **	0.73 **	0.42 *	0.29	−0.06	0.29	−0.48 *	0.41 *
Leaf—2nd season × Leaf—4th season	0.52 **	0.57 **	0.49 **	0.15	0.60 **	0.33	0.19	0.21	0.63 **	−0.13	−0.01
Leaf—3rd season × Leaf—4th season	0.42 *	0.29	0.31	0.53 **	0.64 **	0.32	0.07	0.33	0.20	−0.06	0.53 **

Values highlighted in green and red mean positive and negative significant correlations, respectively (* *p* ≤ 0.05 and ** *p* ≤ 0.01).

**Table 8 plants-10-02661-t008:** Chemical attributes and granulometric fractions of the soil in the experimental area. São Mateus, state of Espírito Santo—Brazil.

Chemical Attributes												
(mg dm^−3^)									(cmol_c_ dm^−3^)			
P ^1^	K ^1^	S	Fe	Zn ^1^	Cu ^1^	Mn ^1^	B	Na	Ca	Mg	Al	H + Al
95	93	6	157	8.4	2.6	14	0.33	7.0	2.0	0.8	0	1.5
pH			V (%)			Organic Matter (dag dm^−3^)						
6.6			67.17			2.1						
Particle size fractions (g kg^−1^)									Texture classification			
Sand			Silt			Clay			Sandy loam			
746			54			200						

P, K, Na, Fe, Zn, Cu, and Mn: Mehlich-1 extractor; pH: in H_2_O; Ca and Mg: extraction: KCI 1 mol L^−1^; Al + H: titration; MO: Embrapa method; V (base saturation); physical analysis: according to Embrapa; clay by the hydrometer method; texture classification, according to the Brazilian Society of Soil Science.

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
