# Peer review of "Concentration of Nutrients in Leaves, Flowers, and Fruits of Genotypes of Coffea canephora"

_plants, 2021, doi:10.3390/plants10122661_

Round 1
Reviewer 1 Report
Dear Autors,
after reviewing your manuscript, we have the following ambiguities that I would like you to clear up
1. which micro and macronutrients are most desirable in particular parts of the plant?
2. what is the relevance of assessing the nutritional value of leaves and flowers, if grains are primarily used?
3. Are further studies planned to what extent the genotype and accumulation of individual compounds affect the quality and taste of grain? In my humble opinion, such an assessment would give a broader picture of the relationships studied here and would affect the actual use of this knowledge in practice.
Reviewer 2 Report
This manuscript presents a study on the macro- and micronutrients concentration in several tissues of nine genotypes of Coffea canephora. In general, the objective of this study is confusing due to the presentation of the results through the text. I think that this manuscript should be wide reorganized and rewritten.
According with the Material and Method section, the authors have used an ANOVA analysis for evaluating their results. However, this ANOVA analysis has not shown in the manuscript. I think that one table with the data of this ANOVA analysis should be included.
In page 2, the authors mention in the text that they use one ‘environmental variation coefficient (CVe)’, but in table 1, this abbreviation (CVe) is associated to one ‘experimental variation coefficient’. Both coefficients are obviously different, and consequently, this should be clarified. On the other hand, they use the notation H2, commonly associated to the broad-sense heritability (relation between genetic variance and phenotypic variance), as ‘genotypic variation coefficient’. This should also be corrected and clarified.
The figures 1-3 are very difficult to visualize, and in many cases represent averages that they are not significant different. These data should be showed in table and joined with the ANOVA table that is missing.
The authors indicate in title of this manuscript that they study the composition of leaves, flowers and fruits, but not the composition of straw. This should be deleted of the manuscript.
The table 2 and 3 are very confusing to understand. They should be clarified. Also, the authors mention in text that they have measured the concentration in four dates, but in tables 2 and 3, they indicated ‘four seasons’. This should be homogenized.
On the other hand, the part of the manuscript about the analysis in leaves should be in front according with the title of manuscript. First, leaves and after flowers and fruits; because this is the remobililization route of these macro- and micronutrients in the plant. For this reason, the leaves show low concentration to the end of process.
The authors should explain as they can calculate the genetic dissimilarity and present one cluster analysis using traits that they are not genes. I would suggest that this analysis would be carried with one principal components analysis (PCA), which would be more correct in this case.
Reviewer 3 Report
General comments
The paper “Concentration of nutrients in leaves, flowers and fruits of genotypes of Coffea canephora” presents data on the macro-and micro-nutrient composition of different plant organs of 9 conilon coffee genotypes at different times and phenological periods. This study sheds light on the variation in plant nutrient composition among different genotypes of conilon coffee that have differing maturity periods. Such a study can provide key information about the nutrient requirements of crops, in this case conilon coffee that may inform plant nutrient management plans. The data presented is interesting and reveals which macro and micronutrients are essential (highly accumulated) in conilon coffee plants. Nevertheless, there are some inconsistencies, contradictions, and ambiguous statements that make the paper unpublishable in its current state. However, these major, as well as some minor issues, can be addressed to allow the paper to be accepted for publication. These are outlined below:
Major corrections:
Abstract
P1L26-27: The last sentence appears to be incorrect since the concentrations of the other major nutrients (i.e. P, Ca, Mg, S etc) were generally higher than the concentration of ‘Fe’. Thus, the authors should revise the sentence to include the names of the different categories i.e. macro and micronutrients to make it clearer and more accurate.
P1L29: The phrase ’due to the mobilization to fruits’ needs to be revised to include the word ‘possibly’ or ‘probably’ before the word ‘due’. This is because mobilization of nutrients to fruits was not directly assessed in this study thus whatever is being proposed as the causative factor for the low nutrient in leaves in the last season is only an assumption and not an evidence-backed fact.
Results
Figures 1 – 3 should be redrawn to include error bars to allow readers to judge the variation in the data and how this informs any statistical significance or otherwise between the genotypes. Furthermore, some of the colors are not easily distinguished from each other. This is especially true for Verdin R and Verdin TA. The authors should consider changing some of these colors.
P3L106: In the last sentence, insert ‘for each nutrient’ before the word ‘do’ to make it clear that comparisons are being made within each nutrient group but not across nutrients. Do the same for Figures 2 – 3.
P4L137-138: The last sentence is not entirely true. Genotype Clementino didn't have the highest concentration of K and Ca. Revise the sentence, by replacing 'all' with 'most'.
Table 2 - 3: Include a grand mean across the genotypes for all the seasons per nutrient to allow for observing the overall trend across the seasons.
Table 2: The second half of the table containing data for Ca, Mg, and S has the Portuguese translation of the word Genotype (i.e Genotipos). Change this to the English version.
P5L174: Add 'for each nutrient' after ‘column’ and before the comma to make it clear that comparisons are only being made per nutrient. Do the same for Table 3 (P7L227).
P5L186: What do the authors mean by ‘a more concentrated reduction’. They should revise the sentence to make it clearer.
P6L215-216: It is not clear what is meant by ‘had a better result isolated in season 3’?
P6L220: The statement ‘Pirata had the highest mean in all seasons’ is not entirely true. Pirata didn’t have the highest in ‘Season 3’.
P7L229-230: This sentence needs to be revised. First off, the word 'stability' is a strong word and cannot be used to describe the data referenced here especially since some had significant differences between genotypes as well as between seasons (even for seasons 1,2, and 4). Secondly, it is not all the genotypes that recorded minor changes in the Zn concentration, some had significant variations. This should be made clearer in the sentence. The authors should also make it clear which data (i.e. between genotypes or seasons) is being described/referenced.
Discussion
The true value of the data presented in this paper lies in its ability to inform plant nutrition management decisions. The authors failed to link and discuss their data in the light of plant nutrition management used in this study and how the results of their study can be used for crop nutrition management. Doing this will improve the relevance of the study and let readers appreciate its potential impact.
The correlation between different nutrients and their significance were never discussed. If the authors want to keep it, then its import should be discussed.
P10L328: The authors write ‘Thus, the results indicate less influence of the environment on nutrient absorption and allocation among the evaluated genotypes…’. However, this statement appears to be contrary to what was expressed earlier (P9L310-311) where the environment (rainfall and temperature) was been touted as contributory factors for the increased nutrient levels. The authors should try to reconcile the 2 seemingly contradictory statements.
P10L354: What do the authors mean by the first sentence. Its meaning is not clear and thus needs to be revised.
P10L355: What do the authors mean by ‘indicate a nutritional diagnosis?’ They need to reword to make the meaning clearer.
P10L358-359: What season is being referred to in the sentence bordering these lines? No mention has been made to any season from the beginning of the paragraph thus the authors need to make the season of interest explicit in the sentence.
P12L428: …’ the parameters already mentioned’… What specific set of parameters are being referred to here? Several parameters have been mentioned including nutritional and environmental parameters. The authors should revise the sentence to make it clearer.
P12L435-436: The authors write, ‘The micronutrient Fe is considered the most accumulated by conilon coffee plants, followed by Mn, B…’ Is this statement in reference to a mean of all plant parts or a specific plant part? This needs to be clarified further, considering the fact that the position of Mn and B are contrary to what had been stated earlier for coffee grains and straw (P11L382).
Materials and methods
P14L507: The authors claim to have used ‘three replicates’. However, it is not known whether these replicates are individual plants or sets of plants in a plot. They should make it more explicit.
P14L521: The author should indicate the month or months in which there was ‘complete maturation’ so that this can be related to the period for leaf sampling. Additionally, since the genotypes had different maturity periods, they should make it clear if sampling was done at one time for all genotypes or based on the maturity period.
P14L533-535: How were these parameters (CVe, CVg and H2) calculated/ determined? (i.e. provide formulae and references).
P14L524-527: This whole paper is centered on the levels of the nutrients listed here. I would expect the authors to provide some more details (or at least a brief description) of the methods/techniques used in determining the concentration of the nutrients.
P15L547-548: This conclusion is contrary to what was stated earlier in the discussion about the impact of the environment on nutrients in flowers (P10L330-332). They need to reconcile these 2 contradictory statements.
Minor comments:
P5L189: Replace ‘is’ with ‘was’
P7L238: Replace ‘are’ with ‘were’.
P7L244: Replace ‘figured’ with ‘featured’.
P7L254: Replace ‘figured’ with ‘featured’.
P8L297: Insert ‘significant’ before ‘correlations’.
P13L484: The ‘2’ in ‘H2O’ should be subscripted whereas the ‘-1’ in ‘mol L-1’ should superscript.
Reviewer 4 Report
Dear Editor,
Dear Dr. Authors,
The research shows practically elaborated characteristics in coffee crops, such as variability in macro and micronutrients among genotypes, which are highly recommended in selecting genotypes with desired well-balanced nutritional value. The problem is that although there are important tweaks in the design, the analysis and results are repetitions of previous work. Not many new methods were learned, and or new lines of coffee weren’t developed. In general, I find the manuscript very clearly written. The purpose, methods, and results are clearly presented.
Overall, this is a good paper.
Some minor revision (typo) only:
Materials and methods pages 14-17. Line 531 and line 536: Correct and make a uniform for significance level for P select one format; with small or upper case or following the format given on Table 5.
Typo in Line 336 ‘tat’, correct.
Line 401-402 correct double square brackets “[4, [30]”.
Round 2
Reviewer 2 Report
I think the authors have addressed most of the problems and reviewed the MS satisfactorily. I positively value the effort made by the authors for the revision of this manuscript.
Minor change
L352. Authors indicated ‘additive genetic variance’. This is incorrect and should be corrected. Authors have evaluated the broad-sense heritability (H2), not the narrow-sense heritability (h2). Consequently they cannot measure the additive genetic variance.
